# Pleasantness Recognition Induced by Different Odor Concentrations Using Olfactory Electroencephalogram Signals

**DOI:** 10.3390/s22228808

**Published:** 2022-11-15

**Authors:** Hui-Rang Hou, Rui-Xue Han, Xiao-Nei Zhang, Qing-Hao Meng

**Affiliations:** 1Tianjin Key Laboratory of Process Measurement and Control, Institute of Robotics and Autonomous Systems, School of Electrical and Information Engineering, Tianjin University, Tianjin 300350, China; 2Tianjin Navigation Instruments Research Institute, Tianjin 300131, China

**Keywords:** pleasantness recognition, olfactory EEG, different odor concentrations, gamma frequency band, power spectral density

## Abstract

Olfactory-induced emotion plays an important role in communication, decision-making, multimedia, and disorder treatment. Using electroencephalogram (EEG) technology, this paper focuses on (1) exploring the possibility of recognizing pleasantness induced by different concentrations of odors, (2) finding the EEG rhythm wave that is most suitable for the recognition of different odor concentrations, (3) analyzing recognition accuracies with concentration changes, and (4) selecting a suitable classifier for this classification task. To explore these issues, first, emotions induced by five different concentrations of rose or rotten odors are divided into five kinds of pleasantness by averaging subjective evaluation scores. Then, the power spectral density features of EEG signals and support vector machine (SVM) are used for classification tasks. Classification results on the EEG signals collected from 13 participants show that for pleasantness recognition induced by pleasant or disgusting odor concentrations, considerable average classification accuracies of 93.5% or 92.2% are obtained, respectively. The results indicate that (1) using EEG technology, pleasantness recognition induced by different odor concentrations is possible; (2) gamma frequency band outperformed other EEG rhythm-based frequency bands in terms of classification accuracy, and as the maximum frequency of the EEG spectrum increases, the pleasantness classification accuracy gradually increases; (3) for both rose and rotten odors, the highest concentration obtains the best classification accuracy, followed by the lowest concentration.

## 1. Introduction

Emotion is an ever-changing psycho-physiological phenomenon that represents the body’s adaptation to environments [1]. As a high-level function of brain activity, emotions affect people’s communication, learning, memory, decision-making, and other mental aspects [2,3,4]. The human exploration of one’s own emotions has never stopped, and emotion has always been the focus of psychology and physiology [5]. Nowadays, with the rapid development of neuroscience, psychology, brain science, computer science, and artificial intelligence, people have deepened their understanding and application of emotions.

At present, for the induction of emotions, researchers generally use visual, auditory, or a combination of visual and auditory stimuli [6,7,8]. However, as one of the most important and original senses, olfaction is naturally emotional, mainly because the olfactory cortex in the brain is directly connected to the limbic system, and the limbic system plays a key role in human emotion and memory [9,10,11]. In fact, olfaction can provide a unique channel for humans to understand their own perceptual world and social-emotional experience, and olfactory stimulation can trigger pleasant or unpleasant emotions, affecting people’s behaviors and decision making. The research of emotion recognition based on olfactory stimulation has important practical significance. For example, in multimedia systems, traditional multimedia systems based on visual and auditory stimuli cannot meet the user’s need for the realism of the experience. Incorporating olfactory into the multimedia system enhances the realism of the user’s emotional experience and satisfies people’s pursuit of high living standards [12]; in medical health, olfactory stimulation can be used as an auxiliary diagnosis of olfactory dysfunction disease and can be used to regulate the mood of patients suffering from depression or mental illness, thus improving the effectiveness of medical treatment [13,14,15].

Many techniques can capture the emotion state under olfactory stimulation [16,17,18,19]. Electroencephalogram (EEG) technology is used in this study because EEG technology is non-invasive, convenient, and inexpensive [20]. Based on EEG technology, much research on the pleasantness classification of olfactory-induced emotions has been performed. To discriminate aromatic olfactory stimuli (standard stimuli) and noisy stimuli, a novel algorithm for evolutionary feature selection and data dimensionality reduction from multiple experimental EEG trials using principal component analysis and a classification method of the recurrent neural network have been developed and verified by Saha et al. [21]. The importance of Saha et al.’s research has been emphasized by the selection of tea-tasters, where the tea-tasters could be ranked by measuring their olfactory perception. Focusing on automatically classifying the pleasant and disgust olfactory stimulation by EEG analysis, the leave-one-subject-out linear discriminant analysis (LDA) classification method is proposed and verified by Lanata et al. [22]. Similarly, the LDA is used to recognize the odor-induced emotions by Kroupi et al. [23]. Their experimental results show that under olfactory stimulation, it is possible to classify pleasant and disgust emotions but that it is difficult to classify pleasant and neutral emotions [23,24]. In addition, the support vector machine (SVM) [25,26,27] and *k*-nearest neighbor (*k*-NN) [28] methods as well as the gamma-band activity are significant for olfactory-induced pleasantness classification.

As shown above, the research on olfactory EEG recognition has made significant progress. However, all the presented studies use only different odor types to stimulate the participants. To the best of our knowledge, the research on emotion recognition caused by different concentrations of odors is currently scarce. For stimulus studies of odor concentration, Tateyama et al. [29] explored the effect of odor concentration on olfactory event-related potential (OERP). The results show that the OERP amplitudes and latencies are associated with the odor concentration. However, this study does not involve emotion research, such as emotion recognition.

This article explores pleasantness recognition induced by different odor concentrations for the first time. The main purpose of this study is (1) to explore the classification possibility of odor concentration-induced emotions using EEG signals, (2) to find the best EEG rhythm wave for this recognition task, (3) to analyze the relationship between recognition accuracy and odor concentrations, and (4) to select the most suitable one among five commonly used classifiers.

The structure of this paper is as follows. In Section 2, the olfactory EEG dataset and methods are illustrated. Section 3 describes the experimental results and Section 4 discusses the results, followed by a conclusion in Section 5.

## 2. Olfactory EEG Dataset and Classification Methods

### 2.1. Experimental Procedures

Two kinds of odors in the T&T olfactometer (from Daiichi Yakuhin Sangyo Co., Ltd., Tokyo, Japan) were used; one smells like roses, termed *rose* odor, and the other is a foul, unpleasant odor, termed *rotten* odor, and each includes five different concentrations. The five concentrations of each odor type from low to high are described as A10^−5.0^, A10^−4.5^, A10^−4.0^, A10^−3.5^, and A10^−3.0^ and C10^−6.0^, C10^−5.5^, C10^−5.0^, C10^−4.5^, and C10^−4.0^, respectively, where the values 10^−3.0^, 10^−3.5^, …, 10^−6.0^ indicate the dilution coefficients of the concentration. For different odor types, the central concentration, e.g., A10^−4.0^ or C10^−5.0^, which corresponds to the detection threshold of normal olfactory sensation, may be different. If someone can smell the central concentration, it means that their sense of smell is normal; otherwise, it is abnormal. Because of the different central concentrations of different odor types, the magnitudes of the exponents (e.g., C10^−6.0^ and A10^−5.0^) of rotten and rose odors are different.

During the experiment, the smells were randomly selected and placed 1–2 cm below the participant’s nose through test papers. Subsequently, a Cerebus system (Blackrock Microsystems, Salt Lake City, UT, USA, Figure 1) was used to collect EEG signals with a 1 kHz sampling rate and a 32-channel EEG cap (according to the international 10–20 system layout). Among the 32 electrodes, 2 electrodes were used as references, and the remaining 30 electrodes were used for data analysis.

There were 13 participants, aged 24.9 ± 2.25 years, who were informed and agreed to the experimental protocol. All the participants in this study are with normal sense of smell, no smell-related diseases, right handedness, and no smoking and drinking habits. For each odor concentration for each participant, 35 stimulations were performed to obtain 35 trials. In each trial, the collection duration of the EEG signal was 10 s, and there were a 4-s interval for two adjacent trials and a 2-min interval for two odor concentrations. Using a 2-min interval, the last odor concentration could be excluded from the experimental room with an exhaust fan, and the pleasantness induced by each concentration was subjectively evaluated. Within each participant, the olfactory EEG data corresponded to a fixed concentration of a certain odor were assigned to one category, with each category containing 35 EEG samples. Thus, for each participant, we obtained 10 (2 kinds of odors ×5 concentrations) categories and 350 (10 × 35) EEG samples. In total, 4550 (350 × 13) EEG samples were collected.

### 2.2. Feature Extraction

The high-frequency noise of EEG signals was removed using a 0.5–70 Hz bandpass filter, and then the power spectral density (PSD) features were calculated in each of the five typical EEG bands, namely, delta (δ): 0.5–3 Hz, theta (θ): 4–7 Hz, alpha (α): 8–13 Hz, beta (β): 14–30 Hz, and gamma (γ): 31–70 Hz.

Figure 2 shows the PSD feature extraction process. First, for each electrode, a Welch periodogram with 100 sample Hamming windows was used to estimate the PSD amplitude. Then, the PSD curves were divided into five segments based on the EEG rhythm.

### 2.3. Classifiers

In the classification stage, we compared five classical and commonly used classifiers, such as SVM [30], naive Bayesian (NB) algorithm [31], *k*-NN algorithm [32], extreme learning machine (ELM) [33], and backpropagation neural networks (BPNN) [34], in which the SVM and BPNN were performed in a MATLAB toolbox.

## 3. Results

### 3.1. Subjective Evaluation Results

In the experiment, each participant performed a subjective evaluation of the pleasure level induced by each odor concentration. For each concentration of each odor, the subjective score ranged from −10 to 10. Positive scores represent pleasant emotions, where greater values represent more pleasant odors; the negative scores represent disgust emotions, where greater absolute values indicated more disgusting odors. The subjective evaluation scores are shown in Table 1.

The results show that different concentrations of odors can induce different degrees of pleasure or disgust. To obtain emotional categories, the average subjective evaluation scores of 13 participants were calculated. As a result, five different levels of pleasant emotions can be induced by the five different concentrations of rose odor, while five different levels of disgust emotions can be induced by the five different concentrations of rotten odor.

To validate the claim of varying levels of emotions with concentrations, statistical tests, specifically, the Friedman significance test, for the subjective evaluation scores of five concentrations of each odor type have been performed, and the significance test results (i.e., *p*-values) for the rose and rotten odors are 4.7 × 10^−3^ and 6.2 × 10^−8^, respectively. Both of these are less than the significance level of 0.05, proving that for the subjective evaluation scores of 13 participants used in this study, the differences between the five concentrations of each odor type can reflect the overall differences between the odor concentrations. In other words, the significance test results prove that the levels of emotion vary with concentrations.

### 3.2. Pleasantness Recognition Induced by Rose Odor Concentrations

In this part, we classified five kinds of pleasant emotions with different levels of pleasure. From Table 1, the average jury scores vary with the five different concentrations of the rose odor, indicating that the degrees of pleasantness induced by the five different concentrations are different. In the training process for pleasantness recognition induced by the rose odor concentrations, five classes correspond to the five concentrations. To verify and compare several classic recognition methods, all the recognition algorithms were trained and tested 10 times by processing the randomly divided training samples (20 samples in each concentration class for each participant) and testing samples (15 samples in each concentration class for each participant).

Quantitatively, the corresponding average classification accuracies across the concentrations of all participants using five classifiers are illustrated in Figure 3, which shows that the SVM with gamma frequency band yields the best classification accuracy, obviously higher than other classification methods. It is also better than the classification accuracy obtained using the full frequency bands (FFB), i.e., using delta, theta, alpha, beta, and gamma frequency bands together. These results suggest that the EEG gamma band is closely related to the olfactory stimulus. Furthermore, Figure 3 shows that as the frequency increases, the accuracy of most classifiers increases, indicating that the olfactory stimulus is highly correlated with the high-frequency band of EEG signals. Specifically, the gamma frequency band yields the best classification accuracy. The accuracy is higher than other frequency bands and outperforms the FFB, indicating the excellent classification ability of the gamma band for different degrees of rose odor-induced pleasantness.

To explore the influence of concentration on recognition accuracy, the average classification accuracies across 13 participants for five different concentrations of the rose odor using the SVM classifier, are shown in Figure 4. For all frequency bands, the highest concentration, i.e., the concentration of A10^−3.0^, obtains the best classification accuracy, followed by the lowest concentration, i.e., the concentration of A10^−5.0^. These results suggest that the highest and lowest concentrations are relatively easy to recognize. The reason may be that the EEG signals induced by the highest and lowest concentrations are different from that induced by other concentrations.

### 3.3. Pleasantness Recognition Induced by Rotten Odor Concentrations

Furthermore, we conducted pleasantness classification induced by the disgusting odor, and we also compared the presented classification methods. In the training process for pleasantness recognition induced by the five concentrations of the rotten odor, five classes correspond to the five concentrations of the rotten odor. Each concentration class contains 35 EEG samples, where 20 samples were randomly selected as the training samples, and the other 15 samples were used to test the training algorithms. The random division of the samples was performed 10 times, and the following results are the average results of the corresponding 10 tests of classifiers.

The comparison results of the five classifiers and five frequency bands are shown in Figure 5. For the classification of disgust emotions induced by the rotten odor, the SVM still produces the best classification accuracy, and the classification accuracy of most classifiers generally increases as the frequency increases. Additionally, similar to the conclusion drawn from Figure 3, Figure 5 shows that for the recognition of rotten odor-induced emotions, the accuracy of the gamma frequency band on each participant is higher than the other frequency bands and outperforms the FFB, indicating the excellent classification ability of the gamma band for different degrees of rotten odor-induced disgust emotions.

Using the SVM classifier, average classification accuracies across 13 participants for five different concentrations of the rotten odor are shown in Figure 6. Similar to Figure 4, the results in Figure 6 suggest that the highest and lowest concentrations are relatively easy to recognize. Similar to the rose odor, the EEG signals induced by the highest and lowest concentrations of the rotten odor are relatively different from those induced by other concentrations.

### 3.4. Emotion Classifications of Pleasure and Disgust

According to the subjective evaluation rules, positive scores represent pleasant emotions, and negative scores represent disgust emotions. We divide emotions into two types: pleasure and disgust. As shown in Table 1, the rose odor corresponds to pleasant emotions, and the rotten odor corresponds to disgust emotions. In other words, five concentrations of the rose odor correspond to pleasant emotion, while five concentrations of the rotten odor correspond to disgust emotion. Subsequently, the corresponding olfactory EEG signals are also classified into two categories for recognition. In the training and testing process for recognition of pleasure and disgust emotions, two classes correspond to two odor types, and each odor type has five concentrations. For each odor type, there are 175 = 35 × 5 EEG samples. The classification results are shown in Figure 7.

Comparing Figure 7 with Figure 3 or Figure 5, it can be seen that the classification accuracy is significantly improved in the two types of emotional classifications of pleasure and disgust, indicating that the emotional classifications of pleasure and disgust are easier to recognize than the different degrees of pleasant or disgust emotions.

In addition, we used the polygon area metric [35] to evaluate the performance of the classifier for emotion classifications of pleasure and disgust. According to [35], we can obtain the calculation formula of the polygon area metric (PAM) as follows:(1)$$\mathrm{PAM}=\frac{\sqrt{3}}{4\times 2.59807}\left(\mathrm{CA}\times SE+SE\times \mathrm{SP}+\mathrm{SP}\times \mathrm{JI}+\mathrm{JI}\times F+F\times \mathrm{AUC}+\mathrm{AUC}\times \mathrm{CA}\right)$$

Considering that Gamma band outperforms other EEG rhythms in the olfactory-induced emotion recognition. Here, we take Gamma band as an example to test PAM performance of each recognition method, and the results are shown in the Table 2, which again illustrates that recognizing the pleasure and disgust emotions is ease and feasible.

## 4. Discussion

### 4.1. Analysis of Outperformance of the Gamma Band and SVM Algorithm

The reason for the outperformance of the gamma band and not the FFB may be that there are redundant features in the FFB. The features extracted from FFB contain the features of delta, theta, alpha, and beta, where the classification ability of these features is weaker than that of the gamma frequency band. More weak classification features may bring redundant features, thereby reducing the classification accuracy.

The reason why SVM outperforms four other classifiers, i.e., NB, *k*-NN, ELM, and BPNN, may be that for the non-massive data, such as the data used in this study, the classification ability of SVM is strong. This is also proved by the experimental results from other researchers [23,24,25,26].

Notably, the analysis of gamma-band in (visual, auditory, etc.) EEG usually has limitations and it is better suited to MEG studies. However, for olfactory EEG, from the perspective of signal recognition, we found that the features of gamma band are better than those of other EEG rhythms, which has also been proved by other scholars [28]. Moreover, we will further analyze the classification value of gamma band, such as extracting and testing different types of features from gamma band, and testing the value of gamma band with more datasets.

### 4.2. Method Design and Novelty

For the recognition methods, the current research only adopted and verified the feasibility of several classical methods. However, this study does not focus on the method innovations, but the innovation of research content, that is, recognizing emotions induced by different odor concentration through EEG analysis. As far as we know, there are few reports about inducing people’s emotions through different odor concentrations and then conducting such emotional recognition through EEG analysis.

Although the experiments in the literature (e.g., [23,24,28]) are similar to those in this study, they are essentially different. For example, most of the experiments in the literature use different odor types to stimulate the participants, rather than stimulating and inducing their emotions with different concentrations of odors.

Considering that the focus of this study is to test the feasibility of recognizing emotions induced by different odor concentrations through EEG, at this stage, we have not compared the adopted methods with those in other papers. In subsequent research, we will design novel recognition algorithms with better performance for this specific recognition tasks, and compare it with other methods in the literature.

Moreover, by considering and comparing our results with previously reported studies in the literature, the novelty of our study is expressed as follows: The detailed contribution points and the main purpose of this study include: (1) to explore the classification possibility of odor concentration-induced emotions using EEG signals, (2) to find the best EEG rhythm wave for this recognition task, (3) to analyze the relationship between recognition accuracy and odor concentrations, and (4) to select the most suitable one among five commonly used classifiers.

### 4.3. Electrodes Corresponding to the Maximum PSD Amplitudes

Taking the gamma frequency band as an example, to explore the maximum of the PSD amplitudes among all electrodes, which is helpful for the determination of the brain regions related to the sense of smell, we tested each odor concentration and obtained the average PSD amplitudes of 35 EEG samples corresponding to each concentration for each participant. The study found that for different concentrations of the same odor type, the electrode corresponding to the maximum of the PSD amplitudes was the same. However, there are differences in the electrodes corresponding to the maximum amplitude of the PSD of the two odor types. Table 3 shows that for most participants, the PSD amplitude of the FP1 or FZ electrodes located in the frontal lobe region of the brain is the maximum. In addition, Kroupi et al. [28] and Alarcao et al. [36] showed that olfactory stimulation is related to frontal and temporal lobes. This may be the reason why the PSD amplitude of the FP1 and FZ electrodes is relatively large under olfactory stimulation.

### 4.4. Limitations

Size of participants: Compared to datasets with a small number of participants, large datasets are more powerful to support the research. The sample size (13 participants) used in our work is small, but we think that the EEG signals collected from 13 participants can also be used to carry out exploratory research of pleasantness recognition induced by different concentrations of odors. Therefore, in this study, the dataset involving 13 participants was presented, which has been openly available on the website [37]. To further analyze the olfactory EEG signals, we are continuing to expand the participants. Next, we will carry out pleasantness recognition induced by odor concentrations on a larger dataset, and look forward to obtaining better results.

Gender non-homogeneity: For the gender ratio of participants, this study involved 13 participants, including 12 men and 1 woman, which is gender non-homogeneity. For this issue, we will increase the number of female participants in the following research, and maintain gender balance while expanding the scale of participants.

### 4.5. Accuracy Analysis with Respect to Sample Size

The training process in our study contained five steps. (1) Randomly select training samples; (2) Extract features from the selected samples; (3) Feed features into the classifier to determine classifier parameters; (4) Use the determined classifier to predict the categories of new EEG samples; (5) Repeat the presented steps 10 times to obtain the average prediction accuracy of 10 times. The results in Figure 3 and Figure 5 were obtained using 20 training samples and 15 testing samples for each odor concentration. To check how the accuracy changes with sample size, we have adjusted the sample size in training and testing processes, and the average classification accuracy of 13 participants is shown in Table 4. The results are obtained using SVM to classify PSD features extracted from gamma band.

Table 4 shows that as the number of training samples increases and the number of testing samples decreases, both the concentration classification accuracies of rose and rotten odors are improved.

### 4.6. Prediction of Concentrations/Emotions by Analyzing EEG Signals

When given the EEG signals, the concentrations/emotions can be effectively predicted. The accuracies shown in Figure 4 and Figure 6 were calculated by comparing the actual concentration/emotion types with the corresponding prediction results, and the prediction results were obtained by processing EEG signals. Figure 4 shows that using the gamma band to predict the five concentrations/emotions of the rose odor, the lowest accuracy value is 91.0%, corresponding to the concentration of A10^−4.0^; Figure 6 shows that using the gamma band to predict the five concentrations/emotions of the rotten odor, the lowest accuracy value is 86.8%, corresponding to the concentration of C10^−5.5^, thus proving that the concentrations/emotions can be effectively predicted.

### 4.7. Odor Selection

At present, this study is only a preliminary exploration of emotion recognition induced by different odor concentrations. At this stage, we have tried to analyze the emotions induced by only pleasant and disgusting odors. Through the preliminary exploration, the feasibility of recognizing olfactory-induced emotions through EEG analysis has been verified.

For recognition of emotions induced by different odor concentrations using EEG signals, the T&T olfactometer which contains five odor types, each with different concentrations, provides a very suitable odor material. After the psychological evaluation, we eliminated the intolerable sweat odor and feces odor, and finally screened out two odor types, one being the pleasant rose odor, the other the disgusting rotten odor, each odor with five different concentrations. The next research will add the analysis of neutral odor-induced emotions according to your valuable comments.

## 5. Conclusions

This study uses EEG technology to recognize pleasantness induced by different concentrations of odors. Using the support vector machine with power spectral density features of the EEG gamma frequency band, the average classification accuracies of 93.55% and 92.24% were achieved for the pleasantness recognition induced by different concentrations of pleasant rose and disgusting rotten odors, respectively. When classifying the pleasant and disgusting emotions, a considerably high accuracy of 99.9% was obtained. The experimental results reveal that (1) it is possible to classify pleasantness emotions induced by odor concentrations; (2) in the EEG rhythm, the gamma band is most related to odor concentration stimuli; (3) compared to the naive Bayesian, *k*-nearest neighbor, extreme learning machine, and backpropagation neural networks, the support vector machine demonstrates the feasibility for recognizing the odor-induced pleasantness.

## Figures and Tables

**Figure 1 sensors-22-08808-f001:**
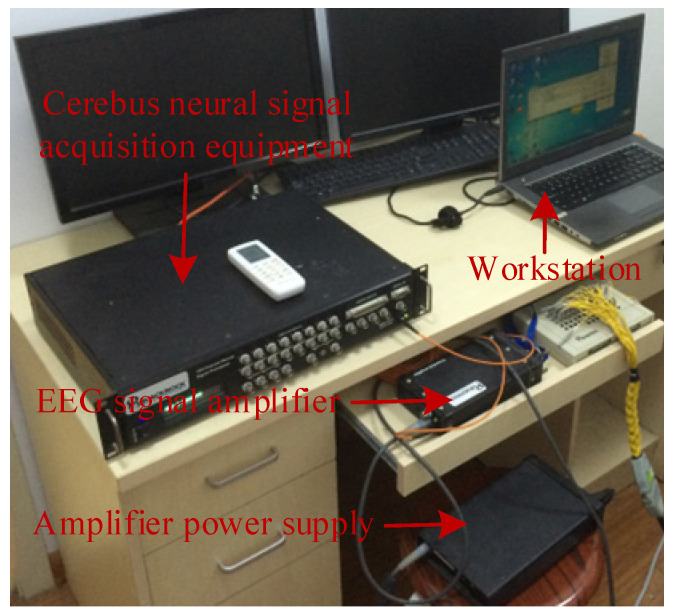
Experimental platform. The experimental platform consists of an amplifier power supply and a signal amplifier, Cerebus signal acquisition equipment, and workstation.

**Figure 2 sensors-22-08808-f002:**
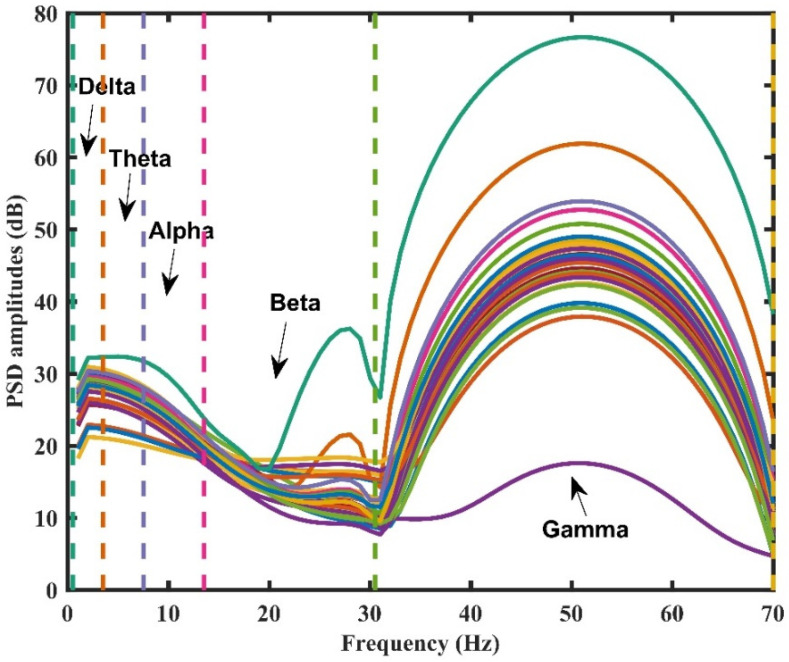
PSD feature extraction of a single participant based on EEG rhythm, which corresponds to a particular concentration of the rose odor, i.e., the A10^−5.0^. Each color corresponds to an EEG electrode. EEG, electroencephalogram; PSD, power spectral density.

**Figure 3 sensors-22-08808-f003:**
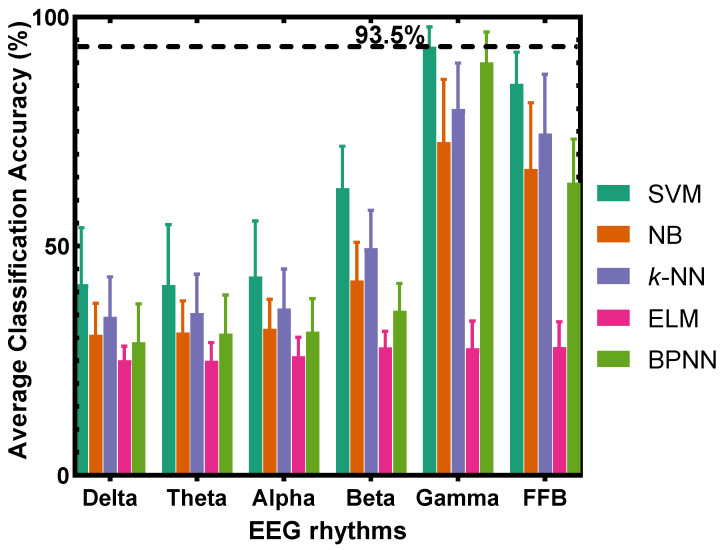
Classification results across concentrations of all participants on the five pleasant emotions induced by the rose odor.

**Figure 4 sensors-22-08808-f004:**
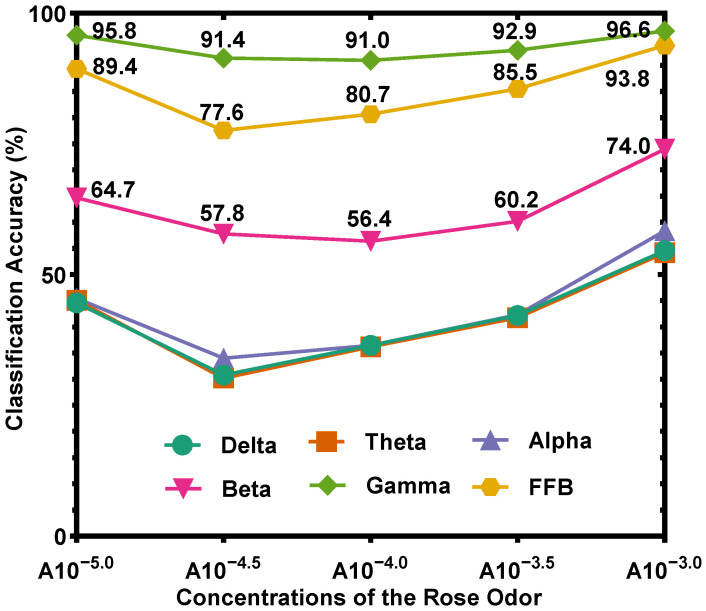
Average classification accuracy across 13 participants for five different rose odor concentrations (using SVM classifier).

**Figure 5 sensors-22-08808-f005:**
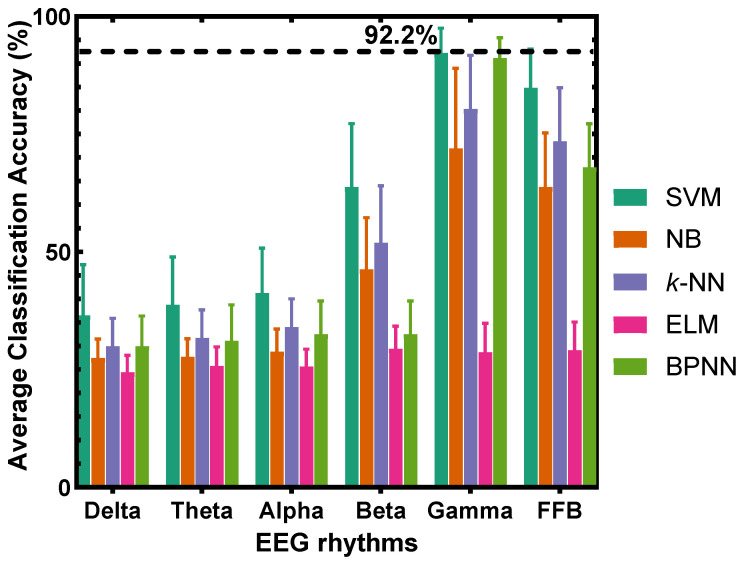
Classification results across concentrations of all participants on the five disgust emotions induced by the rotten odor.

**Figure 6 sensors-22-08808-f006:**
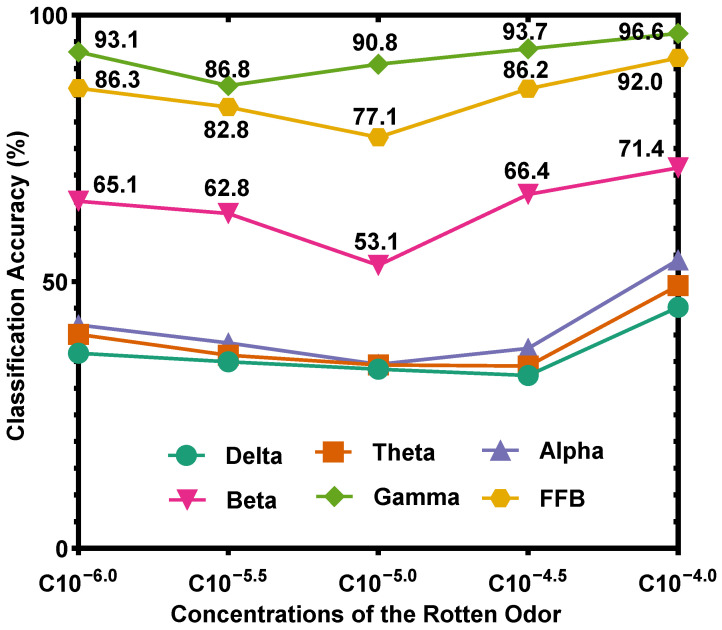
Average classification accuracy across 13 participants for five different rotten odor concentrations (using the SVM classifier).

**Figure 7 sensors-22-08808-f007:**
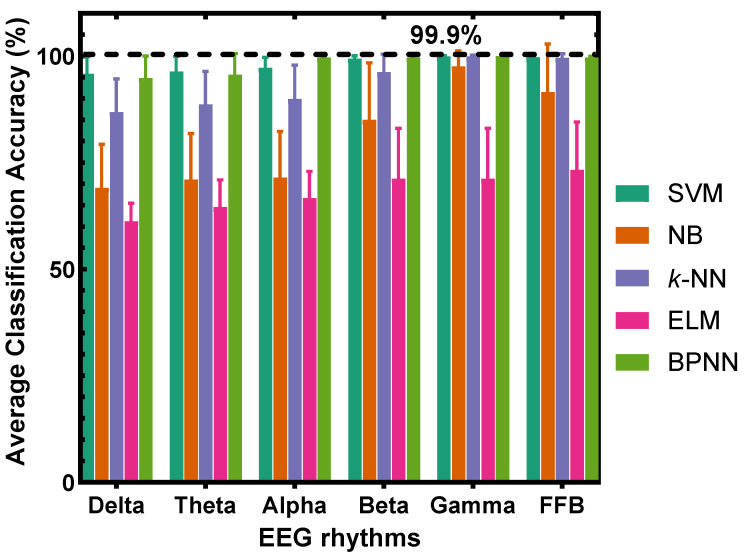
Classification results on two emotions induced by rose and rotten odors. For each odor type, all concentrations were used to determine accuracy.

**Table 1 sensors-22-08808-t001:** The subjective evaluation scores of each participant.

Sub.	Five Different Concentration Levels of Rose (A) and Rotten (C) Odors									
	A10^−5.0^	A10^−4.5^	A10^−4.0^	A10^−3.5^	A10^−3.0^	C10^−6.0^	C10^−5.5^	C10^−5.0^	C10^−4.5^	C10^−4.0^
1 (male)	3	4	1	5	5	−3	−3	−3	−4	−4
2 (male)	2	3	3	4	5	−3	−4	−5	−7	−10
3 (male)	1	1	1	0	0	−5	−6	−7	−8	−10
4 (male)	0	1	1	2	3	−4	−4	−3	−5	−5
5 (male)	2	4	5	6	6	−3	0	−3	−5	−3
6 (male)	5	6	3	7	8	−4	−5	−6	−8	−10
7 (male)	1	1	0	0	1	−1	−2	−2	−3	−5
8 (male)	0	1	1	3	4	−1	−3	−7	−8	−9
9 (male)	4	3	4	7	2	−3	−5	−5	−6	−7
10 (male)	0.5	2	4	5	7	−1	−3	−5	−7.5	−9
11 (male)	0	1	1	2	3	−1	−3	−5	−8	−10
12 (female)	4	5	5	6	7	−2	−6	−4	−5	−9
13 (male)	3	4	2	1	2	−4	−5	−3	−4	−6
Average	1.96	2.77	2.38	3.69	4.08	−2.69	−3.77	−4.46	−6.04	−7.46

**Table 2 sensors-22-08808-t002:** PAM performance comparison of each recognition method.

Methods	SVM	NB	*k*-NN	ELM	BPNN
PAM Value	0.99	0.94	0.99	0.47	0.99

**Table 3 sensors-22-08808-t003:** Electrodes corresponding to the maximum PSD amplitudes of each odor type.

Participants	Rose Odor	Rose Odor
Sub. 1	**FP1**	**FP1**
Sub. 2	**FP1**	**FP1**
Sub. 3	CPZ	CPZ
Sub. 4	**FZ**	**FZ**
Sub. 5	FP2	FP2
Sub. 6	**FP1**	**FP1**
Sub. 7	P3	P3
Sub. 8	**FP1**	**FP1**
Sub. 9	**FP1**	**FP1**
Sub. 10	**FZ**	**FZ**
Sub. 11	**FZ**	**FZ**
Sub. 12	**FP1**	**FP1**
Sub. 13	**FP1**	**FP1**

**Table 4 sensors-22-08808-t004:** Results of accuracy change with sample size in training and testing processes. Take the average classification accuracy of 13 participants as an example. The results are obtained using the SVM to classify PSD features extracted from the gamma band. The percentages represent the average classification accuracies across the concentrations.

Training Sample Size for Each Concentration	Testing Sample Size for Each Concentration	Average Classification Accuracy (%) of Five Concentrations of Rose odor	Average Classification Accuracy (%) of Five Concentrations of Rotten Odor
5	30	75.5	75.8
10	25	87.6	86.9
15	20	90.9	91.5
20	15	93.5	92.2
25	10	94.9	95.6
30	5	96.0	96.7

## Data Availability

The raw data used throughout the paper are openly available on the website [37].

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
