# Peer review of "Pleasantness Recognition Induced by Different Odor Concentrations Using Olfactory Electroencephalogram Signals"

_sensors, 2022, doi:10.3390/s22228808_

Round 1
Reviewer 1 Report
The paper is well-designed and readable. However, I have some shortcomings which are given below:
The proposed feature extraction and classification methods do not contain any novelty, which are commonly used methods in EEG based classification research. Additionally, the experiment also analyzed by many researcher in the literature. So, it would be better to compare your results with the reported results in the literature. Moreover, the authors should express their novelty(s) by considering previously reported studies in the literature.
The authors should use newly proposed polygon area metric in order to evaluate the performance of the classifier at least for two-class problems like emotion classifications of pleasure and disgust. More information can be found in the following works:
· https://www.mathworks.com/matlabcentral/fileexchange/74136-polygon-area-metric-for-classifier-evaluation#:~:text=This%20study%20proposes%20a%20stable,the%20need%20for%20several%20metrics.
In table 1, there are 0 values. How did the authors consider these values, why?
Author Response
Comments:
The paper is well-designed and readable. However, I have some shortcomings which are given below:
Q1: The proposed feature extraction and classification methods do not contain any novelty, which are commonly used methods in EEG based classification research. Additionally, the experiment also analyzed by many researcher in the literature. So, it would be better to compare your results with the reported results in the literature. Moreover, the authors should express their novelty(s) by considering previously reported studies in the literature.
Response to Comments No. 1: Thank you very much for the important comment. We agree that for the recognition methods, the current research only adopted and verified the feasibility of several classical methods. However, this study does not focus on the method innovations, but the innovation of research content, that is, recognizing emotions induced by different odor concentration through EEG analysis. As far as we know, there are few reports about inducing people’s emotions through different odor concentrations and then conducting such emotional recognition through EEG analysis.
Although the experiments in the literatures (e.g. the following [1-3]) are similar to those in this study, they are essentially different. For example, most of the experiments in the literatures use different odor types to stimulate the participants, rather than stimulating and inducing their emotions with different concentrations of odors.
Considering that the focus of this study is to test the feasibility of recognizing emotions induced by different odor concentrations through EEG, at this stage, we have not compared the adopted methods with those in other literatures. According to your valuable comments, we will design novel recognition algorithms with better performance for this specific recognition tasks, and compare it with other methods in the literatures.
Moreover, by considering and comparing with previously reported studies in the literature, the novelty of our study is expressedd as follows.
The detailed contribution points and the main purpose of this study include: 1) to explore the classification possibility of odor concentration induced emotions using EEG signals, 2) to find the best EEG rhythm wave for this recognition task, 3) to analyze the relationship between recognition accuracy and odor concentrations, and 4) to select the most suitable one among five commonly used classifiers.
The above explanations have been added in the Section 4.2 of Discussion of the revised manuscript.
[1] Kroupi, E.; Vesin, J. M.; Ebrahimi, T. Subject-independent odor pleasantness classification using brain and peripheral signals, IEEE Trans. Affect. Comput. 2016, 7, 422–434.
[2] Kroupi, E.; Yazdani, A.; Vesin, J. M.; Ebrahimi, T. EEG correlates of pleasant and unpleasant odor perception, ACM Trans. Multimedia Comput., Commun. Appl. 2014, 11, 1-17.
[3] Aydemir, O. Olfactory recognition based on EEG gamma-band activity, Neural Comput. 2017, 29, 1667–1680.
Q2: The authors should use newly proposed polygon area metric in order to evaluate the performance of the classifier at least for two-class problems like emotion classifications of pleasure and disgust. More information can be found in the following works:
https://www.mathworks.com/matlabcentral/fileexchange/74136-polygon-area-metric-for-classifier-evaluation#:~:text=This%20study%20proposes%20a%20stable,the%20need%20for%20several%20metrics.
Response to Comments No. 2: Thank you very much for the valuable suggestion. According to your suggestion, we have used the polygon area metric to evaluate the performance of the classifier for emotion classifications of pleasure and disgust in Section 3.4 of the revised manuscript. Also, the related reference has been added in the revised manuscript.
Considering that Gamma band outperforms other EEG rhythms in the olfactory induced emotion recognition. Here we take Gamma band as an example to test PAM performance of each recognition method, and the results are shown in the following Table 1, which again illustrates that recognizing the pleasure and disgust emotions is ease and feasible.
Table 1. PAM performance comparision of each recognition method
Methods |
SVM |
NB |
k-NN |
ELM |
BPNN |
PAM Value |
0.99 |
0.94 |
0.99 |
0.47 |
0.99 |
[4] Aydemir, O. A new performance evaluation metric for classifiers: polygon area metric, J. Classif. 2021, 38, 16-26.
Q3: In table 1, there are 0 values. How did the authors consider these values, why?
Response to Comments No. 3: Thank you very much for the valuable comment. The appearance of 0 values is normal, mainly because of the difference of participants’ olfactory sensitivity, and the individual difference of sensory evaluation. During the experiments, we asked participants to give scores according to their real experience and feel. So, although the scores of different participants vary greatly, they are reasonable and consistent with the actual situation.

Reviewer 2 Report
Authors address a topic of high interest, namely, the recognition of olfactory-induced emotion using EEG. The authors use an original approach. Unfortunately their results do not have sufficient evidence to support their hypotheses. Detailed comments below.
1) It’s not easy to understand why authors used for analysis only pleasant and disgust odors and excluded the neutral odors
2) The motivation why the odors were selected (pilot psychometric studies and etc.) should be added.
3) The analysis of gamma-band in EEG usually has limitations and better suits to MEG studies. How did you solve the well-known problems in analyzes of EEG data in the gamma range?
4) The most important limitation of the study – low amount of low-information participants (no gender, no questionnaires or subjective self-assessments)
Author Response
Comments:
Authors address a topic of high interest, namely, the recognition of olfactory-induced emotion using EEG. The authors use an original approach. Unfortunately their results do not have sufficient evidence to support their hypotheses. Detailed comments below.
Q1: It’s not easy to understand why authors used for analysis only pleasant and disgust odors and excluded the neutral odors
Response to Comments No. 1: Thank you very much for the valuable comment. At present, this study is only a preliminary exploration of emotion recognition induced by different odor concentrations. At this stage, we have tried to analyze the emotions induced by only pleasant and disgust odors. Through the preliminary exploration, the feasibility of recognizing olfactory induced emotions through EEG analysis has been verified. The next research will add the analysis of neutral odor induced emotions according to your valuable comments.
The above descriptions have been added in the Section 4.6 of the revised manuscript.
Q2: The motivation why the odors were selected (pilot psychometric studies and etc.) should be added.
Response to Comments No. 2: Thank you very much for the important comment. For recognition of emotions induced by different odor concentrations using EEG signals, the T&T olfactometer which contains five odor types, each with different concentrations, provides a very suitable odor material. After the psychological evaluation, we eliminated the intolerable sweat odor and feces odor, and finally screened out two odor types, one is the pleasant rose odor, the other is the disgusting rotten odor, each odor with five different concentrations.
The above descriptions have been added in the Section 4.6 of the revised manuscript.
Q3: The analysis of gamma-band in EEG usually has limitations and better suits to MEG studies. How did you solve the well-known problems in analyzes of EEG data in the gamma range?
Response to Comments No. 3: Thank you very much for the important comment. We agree that, the analysis of gamma-band in (visual, auditory, etc.) EEG usually has limitations and it is better suits to MEG studies. However, for olfactory EEG, from the perspective of signal recognition, we found that the features of gamma band are better than those of other EEG rhythms, which has also been proved by other scholars (the following [1], i.e. the [28] in the manuscript). Moreover, we will further analyze the classification value of gamma band, such as extracting and testing different types of features from gamma band, and testing the value of gamma band with more data sets.
[1] Aydemir, O. Olfactory recognition based on EEG gamma-band activity, Neural Comput. 2017, 29, 1667–1680.
The above descriptions have been added in the Section 4.1 of the revised manuscript.
Q4: The most important limitation of the study – low amount of low-information participants (no gender, no questionnaires or subjective self-assessments)
Response to Comments No. 4: Thank you very much for the important comment. All the participants in this study are with normal sense of smell, no smell-related diseases, right handedness, no smoking and drinking habits. Related information has been added to Section 2.1, and the gender information has been added to Table 1 of the revised manuscript.
For the size of participants, considering this study is a preliminary exploration of emotion recognition induced by different odor concentrations, and the main purpose is to explore the classification possibility of odor concentration induced emotions using EEG signals, the current 13 participants were used to verify this feasibility. The related explanations have been described in Section 4.3 of the priviously submitted manuscript as follows.
“Compared to data sets with a small number of participants, large data sets are more powerful to support the research. The sample size (13 participants) used in our work is small, but we think that the EEG signals collected from 13 participants can also be used to carry out exploratory research of pleasantness recognition induced by different concentrations of odors. Therefore, in this study, the data set involving 13 participants was presented, which has been openly available on the website [37]. To further analyze the olfactory EEG signals, we are continuing to expand the participants. Next, we will carry out pleasantness recognition induced by odor concentrations on a larger data set, and look forward to obtaining better results.”

Round 2
Reviewer 1 Report
The authors addressed all the shortcomings. I believe that this version is has a potential for publication.
Author Response
Comments:
The authors addressed all the shortcomings. I believe that this version is has a potential for publication.
Response: Thank you very much for the positive comment.

Reviewer 2 Report
The section "limitations" should be added, where the few number of participants and gender non-homogeneity should be notified and discussed
Author Response
Comments:
The section “limitations” should be added, where the few number of participants and gender non-homogeneity should be notified and discussed.
Response: Thank you very much for the valuable comment. The Section “limitations” (Section 4.3) has been added, where the few number of participants and gender non-homogeneity have been notified and discussed as follows.
“Size of participants: Compared to data sets with a small number of participants, large data sets are more powerful to support the research. The sample size (13 participants) used in our work is small, but we think that the EEG signals collected from 13 participants can also be used to carry out exploratory research of pleasantness recognition induced by different concentrations of odors. Therefore, in this study, the data set involving 13 participants was presented, which has been openly available on the website [37]. To further analyze the olfactory EEG signals, we are continuing to expand the participants. Next, we will carry out pleasantness recognition induced by odor concentrations on a larger data set, and look forward to obtaining better results.
Gender non-homogeneity: For the gender ratio of participants, this study involved 13 participants, including 12 men and one woman, which is gender non-homogeneity. For this issue, we will increase female participants in the following research, and maintain gender balance while expanding the scale of participants.”
